# Inheritance bias of deletion-harbouring mtDNA in yeast: The role of copy number and intracellular selection

Nataliia D. Kashko[1], Georgii Muravyov[1], Iuliia Karavaeva[1], Elena S. Glagoleva[2,3], Maria D. Logacheva[3,4], Sofya K. Garushyants[1¤], Dmitry A. Knorre[5]*

**1** Faculty of Bioengineering and Bioinformatics, Lomonosov Moscow State University, Moscow, Russia, **2** Faculty of Biology, Lomonosov Moscow State University, Moscow, Russia, **3** Vavilov Institute of General Genetics Russian Academy of Sciences, Moscow, Russia, **4** Center for Molecular and Cellular Biology, Moscow, Russia, **5** A.N. Belozersky Institute of Physico-Chemical Biology, Lomonosov Moscow State University, Moscow, Russia

¤ Current address: Division of Intramural Research, National Library of Medicine, National Institutes of Health, Bethesda, Maryland, United States of America
* knorre@belozersky.msu.ru

## Abstract

During sexual reproduction, fungi usually inherit mtDNA from both parents, however, the distribution of the mtDNA in the progeny can be biased toward some mtDNA variants. For example, crossing *Saccharomyces cerevisiae* strain carrying wild type (*rho⁺*) mtDNA with the strain carrying mutant mtDNA variant with a large deletion (*rho⁻*) can produce up to 99–100% of *rho⁻* diploid progeny. Two factors could contribute to this phenomenon. First, *rho⁻* cells may accumulate more copies of mtDNA molecules per cell than wild-type cells, making *rho⁻* mtDNA the prevalent mtDNA molecule in zygotes. This consequently leads to a high portion of *rho⁻* diploid cells in the offspring. Second, *rho⁻* mtDNA may have a competitive advantage within heteroplasmic cells, and therefore could displace *rho⁺* mtDNA in a series of generations, regardless of their initial ratio. To assess the contribution of these factors, we investigated the genotypes and phenotypes of twenty two *rho⁻* yeast strains. We found that indeed *rho⁻* cells have a higher mtDNA copy number per cell than *rho⁺* strains. Using an *in silico* modelling of mtDNA selection and random drift in heteroplasmic yeast cells, we assessed the intracellular fitness of mutant mtDNA variants. Our model indicates that both higher copy numbers and intracellular fitness advantage of the *rho⁻* mtDNA contribute to the biased inheritance of *rho⁻* mtDNA.

## Authors summary

Eukaryotic cells contain multiple copies of mitochondrial DNA (mtDNA), which encode essential mitochondrial proteins required for cell functioning. However,

**Data availability statement:** All data are present within the manuscript and supplementary materials. Short-read sequencing results of *rho⁻* strains are available in the SRA archive: BioProject PRJNA1039586, SRA study SRP471445. Julia language script for simulation of mtDNA drift in yeast clonal populations is available under the link: https://github.com/dknorre/YeastHeteroplasmy

**Funding:** This work was supported by the Russian Science Foundation (project 22-14-00108, https://rscf.ru/en/). Grant was received by D.A.K. The funders had no role in study design, data collection and analysis, decision to publish, or preparation of the manuscript.

**Competing interests:** The authors have declared that no competing interests exist.

some mtDNA molecules can acquire mutations – such as deletions – that eliminate protein-coding genes. Although these defective mtDNA molecules cannot sustain mitochondrial activity, certain mutant variants are thought to replicate more efficiently than functional ones. Over time, these mutants may outcompete functional mtDNA and dominate the cellular mtDNA pool. In this study, we used baker's yeast as a model system to investigate how functional and mutated mtDNAs compete within cells. Yeast cells inherit mitochondria from both parental cells, making it possible to generate cells with mixed mtDNA types. We isolated and analysed 22 yeast strains, each carrying a different mutant mtDNA variant, and assessed how effectively these mutant mtDNAs can displace functional mtDNA. By measuring cell traits, such as growth rate and mtDNA copy number, and performing computer simulations, we discovered that some mutant mtDNAs gain a replicative advantage despite being detrimental to the cell. Moreover, we found that secondary deletions frequently occur following the initial deletion. These results shed light on the dynamics of mtDNA competition within cells and offer potential implications for understanding mitochondrial evolution.

## Introduction

Eukaryotic cells usually contain multiple not necessarily identical copies of mitochondrial DNA (mtDNA). This phenomenon is referred to as mitochondrial heteroplasmy. Heteroplasmy is linked to a variety of genetic diseases [1] and aging [2]. The presence of multiple mtDNA molecules within cells establishes a complex hierarchy of mtDNA populations. This hierarchy enables natural selection to operate at various levels of biological organisation, such as within individual cells and organisms, and between organisms [3–6]. As a result, "selfish" mtDNA variants may gain an advantage within individual cells, despite being neutral or exhibiting deleterious effects at the cellular or organismal levels [5,7–9].

Several experimental observations support the idea that mtDNA selection acts on different levels of organisation. First, pathogenic mitotypes containing mutations in the D-loop, a non-coding region of mtDNA controlling the initiation of its replication [10], gradually replace donor mtDNA during the development of a multicellular organism from an embryo [11]. Second, the common starling population in Australia was shown to carry two coexisting mtDNA variants (mitotypes). One of the mitotypes demonstrated a moderate advantage at the intracellular level while decreasing the fitness of the whole organism [12]. Next, natural isolates of nematodes *Caenorhabditis briggsae* harbour two distinct mtDNAs: a full-length variant and a deleterious Δ*nad5* variant lacking a portion of the *NAD5* gene. The proportion of Δ*nad5* mtDNA increased across generations if the populations underwent frequent bottlenecks [13]. Finally, it has been shown that in the baker's yeast *Saccharomyces cerevisiae*, the selection of mtDNA also depends on population size. In an experimental evolution study, a deleterious mtDNA variant displaced the parental wild-type variant in small populations, but not in the larger ones [14].

PLOS Genetics

S. cerevisiae has been proven to be an invaluable model for exploring intracellular mtDNA selection. This yeast species can proliferate under fermentative conditions in the presence of mtDNA mutations that disrupt oxidative phosphorylation (referred to as the rho⁻ genotype) or in the complete absence of mtDNA (rho⁰ genotype). Laboratory yeast strains often produce rho⁻ mutants, i.e., having rho⁻ genotype of their mtDNA. Such mutants typically contain large deletions in their mtDNA sequences but usually retain at least one of the eight replication origins [15–17]. Some of these replication origins, namely ORI2, ORI3 and ORI5, are reported to be more active than others [18]. Meanwhile, yeast cells with rho⁻ and rho⁰ genotypes can be differentiated from the wild type (rho⁺) cells by their 'petite' phenotype. This phenotype is defined by slow growth and an inability to utilise non-fermentable carbon sources [19]. The growth rate of petite cells is also reduced because they have only limited capacity to synthesise some amino acids [20]. Additionally, S. cerevisiae has biparental mtDNA inheritance, where both gametes (mating haploid cells) transmit their mtDNAs to the diploid progeny [21], which allows easy production of heteroplasmic yeast cells.

Intriguingly, in the case of some rho⁻ mitotypes, the rho⁻ X rho⁺ cell crossings produce mainly petite diploid cells [22]. This phenomenon, known as mtDNA suppressivity, suggests that rho⁻ mtDNAs might be positively selected, even though it would cause a detrimental (petite) phenotype. On one side, rho⁻ mtDNAs could have a replication rate advantage in heteroplasmic cells. Indeed, some rho⁻ mtDNAs incorporate radioactively labelled nucleotides faster than the rho⁺ full-length molecules [23]. On the other side, rho⁻ haploid strains typically contain a significantly higher quantity of mtDNA molecules per cell [24]. The increased number of mtDNA molecules in rho⁻ cells compared to rho⁺ cells may explain the preferential inheritance of rho⁻ mitotypes in rho⁻ x rho⁺ crossings, even in the absence of intracellular selection. However, the relative contribution of these factors to mtDNA suppressivity remains not fully characterised.

In this study, we generated a set of 22 rho⁻ S. cerevisiae strains, mapped deletions in their mtDNA and analysed mtDNA copy number per cell and suppressivity for each mutant. To assess mtDNA copy number contribution to the degree of suppressivity, we simulated mtDNA selection and genetic drift during crossings using a stochastic model. Utilising this model, which incorporates estimates of mtDNA copy numbers per cell and growth rates of petite cells, we calculated possible suppressivity values under the assumptions of neutral drift and varying levels of replication advantage for rho⁻ mtDNAs. Together, our experimental data and simulations support the assumption that intracellular selection favours rho⁻ mtDNAs in heteroplasmic yeast cells.

## Results

### mtDNA copy number in rho⁻ strains correlates with their suppressivity

To identify the genomic features that contribute to suppressivity, we generated 23 spontaneous rho⁻ strains from a parental haploid strain [mat alpha rho⁺ URA⁺ leu⁻]. To measure their suppressivity, we crossed those strains with the rho⁺ strain of the opposing mating type [mat a rho⁺ ura⁻ LEU⁺]. We defined the suppressivity of the rho⁻ strains as the percentage of diploid progeny cells that are unable to grow on glycerol (Fig 1A). One of the low-suppressive strains was found to be rho⁰ and was therefore excluded from further analysis. The suppressivity of the remaining strains ranged from 19.4 to 90.2 percent (S1 Table).

Next, we isolated total DNA from the rho⁻ strains and performed whole-genome sequencing to identify the position of deletions in their mtDNA. We show that the obtained rho⁻ mtDNA mutants vary in the deletion length and position (Fig 1B). In 7 out of 22 strains, the position of the primary deletion was confirmed by Sanger sequencing (S1 Table). We found that the studied strains retained 76.8 to 14.8 percent of the unique parental mtDNA sequence. All strains retained at least one of the eight predicted replication origins (Fig 1B). We also sequenced a previously isolated hypersuppressive (HS) rho⁻ strain which showed on average 99% suppressivity and retained only a short 2.2 kb fragment of mtDNA spanning a region flanked by ORI2 and ORI7 [25]. In our set of strains, we did not find a strong association between the retained replication origins and suppressivity, however, it may be noticed that two low-suppressive strains, namely IIa10 and 13, lack all three highly active replication origins: ORI2, ORI3 and ORI5 (Fig 1B).

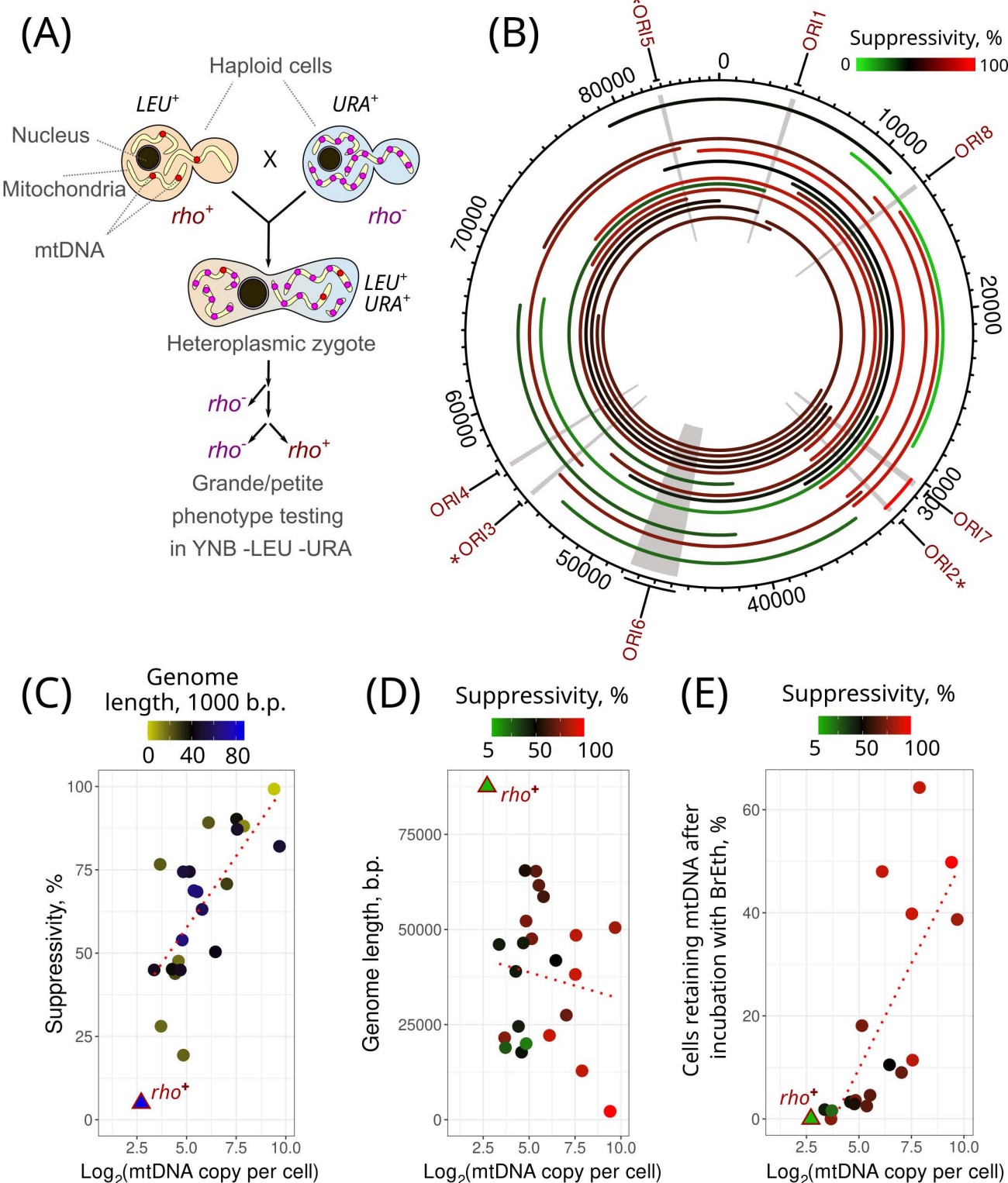

**Fig 1. Suppressivity of spontaneous *rho⁻* strains.** (A) Experimental design for estimating suppressivity; (B) *S. cerevisiae* mitochondrial DNA map featuring the locations of replication origins. The arcs on the map illustrate the mtDNA segments preserved in the spontaneous *rho⁻* mutants examined

in this study. The colour of the arcs corresponds to the average suppressivity of the respective *rho⁻* strains. The arcs are sorted from the longest to the shortest *rho⁻* mitochondrial genomes, with the longest being the closest to the center. Replication origins with higher activity are marked with asterisks (*). (C) Correlation of *rho⁻* mtDNA copy number per nuclear genome estimated from the NGS data with suppressivity (Kendall's rank correlation tau = 0.471, p-value = 0.001); (D) *rho⁻* mtDNA genome size does not show significant correlation with mtDNA copy number per nuclear genome (Kendall's rank correlation tau = 0.105, p-value = 0.45); (E) The ability of *rho⁻* yeast cells to retain mtDNA upon growth with the DNA-intercalating agent ethidium bromide (BrEth) correlates with *rho⁻* mtDNA copy number (Kendall's rank correlation tau = 0.72, p-value = 2×10⁻⁵). Data for the *rho⁺* strain is shown in all plots but is not included in the linear model and correlation tests.

To assess the relative mtDNA copy number for each *rho⁻* strain, we compared the average read coverage depth for mtDNA with the one for the nuclear genome (nDNA, S2 Table). However, because the yeast mitochondrial genome contains regions with very high AT content, these AT-rich sequences pose difficulties during sequencing and read mapping. As a result, mapping algorithms map fewer reads to AT-rich regions, which leads to the non-uniform read coverage along mtDNA (S1 Fig). To overcome this issue, other studies routinely assess mtDNA copy number only in the region of high GC-content spanning from position 14,000–20,000 [26], where read depth is indeed uniform (S1 Fig). Unfortunately, this region was absent in some of our *rho⁻* strains. Thus, we selected three short mtDNA regions (coordinates 8002–8153, 31222–31305, 48195–48296, based on the reference yeast mitochondrial genome) to ensure that each *rho⁻* strain retains at least one of these regions (S2 Table). Notably, these regions had the read depth 2.3-3.4 fold lower than the read depth of the GC-rich 14,000–20,000 region, suggesting that the actual mtDNA copy number estimate using these regions could be underestimated by several fold (S2 Table).

To estimate the mitochondrial DNA copy number in our *rho⁻* strains, we estimated the copy number for each of the available regions and selected the highest value among them. We validated results obtained from the procedure described above for a subset of *rho⁻* strains by quantitative PCR with three pairs of primers designed for the regions used for the NGS copy number assessment (S3 Table and S1 Fig). NGS and qPCR approaches for quantification of mtDNA/nDNA ratios showed concordant results, the values were positively correlated with Kendall's rank correlation tau = 0.683, p-value = 8.2×10⁻⁵ (S2 Fig). However, the absolute values of the mtDNA/nDNA ratios obtained with NGS and qPCR methods substantially differed (see S2 and S4 Tables). For instance, according to NGS, the mtDNA copy number of the control wild-type strain ranged from 3.0 to 15.5 copies per haploid nuclear genome, depending on the mtDNA region selected as reference (S2 Table). In contrast, qPCR provided a higher estimate of 22.2-26.6 mtDNA copies per nuclear genome (S4 Table), which aligns better with the estimates reported in other studies [27–29].

We used the NGS-based evaluation of the mtDNA copy number and correlated it with the corresponding strain's suppressivity. We showed that mtDNA/nDNA ratios of the *rho⁻* strains positively correlate with their average suppressivity (Figs 1C and S3, Kendall's rank correlation tau = 0.47, p-value = 0.001). At the same time, we found no correlation between the length of the mtDNA remaining in the cells and mtDNA copy number (Fig 1D) or suppressivity (S4 Fig). Since both methods for determining the mtDNA/nDNA ratio (qPCR-based and NGS-based) relied on DNA amplification, we decided to complement those measurements with another approach. We hypothesised that *rho⁻* strains with increased mtDNA copy number would be more resilient to the loss of mtDNA in response to the inhibition of mtDNA replication. To test this, we treated *rho⁻* strains with the DNA-intercalating agent, ethidium bromide. Then, we stained the cells with DAPI to visualise nuclear and mitochondrial DNA in the cells. Our findings confirmed that *rho⁻* cells with high mtDNA/nDNA ratios were more resistant to mtDNA loss (Fig 1E). In the analysed set of *rho⁻* strains, the percentage of *rho⁻* cells retaining mtDNA after incubation with ethidium bromide correlated with the average mtDNA/nDNA ratios in the strain (Fig 1E, Kendall's rank correlation tau = 0.72, p-value = 2×10⁻⁵) and suppressivity (Kendall's rank correlation tau = 0.56, p-value = 8×10⁻⁴, S5 Fig). Taken together, three independent approaches to estimate mtDNA copy number show a positive correlation between the mtDNA copy number and suppressivity.

## Suppressivity drifts after genetic bottleneck

To explore the effects of genetic drift and selection on *rho⁻* mtDNA variants in yeast cell populations, we investigated whether suppressivity remains constant in *rho⁻* strains or is subject to drift. To test this, we took several strains with

varying suppressivity from our set and derived subclones by selecting colonies that presumably originated from a single cell. We evaluated the suppressivity of these subclones as described earlier. Fig 2A shows that the suppressivity of a parental strain that did not undergo a bottleneck by a subclone selection remained stable in independent experiments. At the same time, some subclones obtained from single cells showed similar suppressivity, which, however, strongly differed from the average suppressivity of the parental *rho⁻* strain. This suggests the existence of secondary mtDNA mutations in *rho⁻* strains that affect their suppressivity. To test this possibility, we sequenced ten subclones of two *rho⁻* strains, specifically *IIc11* and *Ia14*. We observed that all subclones of *IIc11* had lost an additional various portions of mtDNA, whereas in the case of *Ia14*, one subclone exhibited a large secondary deletion. The other clones showed similar coverage maps, with only a small portion of mtDNA lost in subclones 1 and 3 (Fig 2B).

Our experiments demonstrated that *rho⁻* strains can alter their suppressivity following a genetic bottleneck, prompting us to investigate whether there is a consistent trend towards increased or decreased suppressivity. To examine this, we performed a post hoc analysis of the suppressivity of 76 *rho⁻* strains previously obtained in our laboratory. These strains include the *rho⁻* strains sequenced in this study, as well as additional spontaneous *rho⁻* strains for which multiple suppressivity assessments were available. They include other spontaneous *rho⁻* strains originated from the same *W303 URA⁺ mat alpha* parental *rho⁺* strain which were not selected for sequencing, and *rho⁻* strains obtained from different parental strains with the *W303* background. The initial assessment was conducted immediately after the narrow bottleneck, following the isolation of each *rho⁻* strain (S6 Fig). Subsequent analyses were carried out after an additional (non-controlled) number of generations. We found that the initial suppressivity assessment was on average lower ($5.9\% \pm 10.9\%$) than the subsequent ones (S6 Fig), although there were some isolates whose initial assessments were higher than the subsequent ones. Wilcoxon signed-rank test with continuity correction rejected the hypothesis that the medians of first and subsequent assessments are equal (p-value $= 9.9 \times 10^{-6}$). At the same time, it might be noticed that suppressivity increased to a higher extent in the strains with low initial suppressivity (S6 Fig).

### *rho⁻* mtDNA suppressivity does not affect the strain growth rate

In our experimental setup for the suppressivity assessment, variation in *rho⁻* strain suppressivity could be influenced by the difference in growth rates. For example, if a highly suppressive *rho⁻* strain has a higher proliferation rate than a low suppressivity *rho⁻* strain, diploid strains with homoplasmy of the highly suppressive *rho⁻* variant can undergo more divisions than the corresponding low suppressivity *rho⁻* strain. As a result, it would produce more *petite* cells in the suppressivity assay. To test this possibility, we measured the maximal growth rate of our *rho⁻* and *rho⁺* strains, as well as *rho⁰* cells as a control. We found no pronounced differences in growth rate between *rho⁻* and *rho⁰* strains, while the *rho⁺* strain expectedly displayed higher maximal growth rate (Fig 3A, 3B and S5 Table). As the conditions during crossings deviate from those in an exponentially growing yeast suspension, we additionally tested the change in the number of *URA⁺* cells (colony-forming units) before and after the crossing experiments (Fig 3C). Surprisingly, there was no difference in growth rates of *rho⁺* and *rho⁻/rho⁰* strains under these conditions, as the number of *URA⁺* cells increased approximately tenfold regardless of the mtDNA type for all tested strains/crossings (Fig 3D).

We reasoned that the similar growth rates observed between *rho⁺* and *rho⁻/rho⁰* strains under crossing conditions stem from the inhibition of *rho⁺* cell growth by the mating pheromone (alpha factor), which is secreted by cells of the opposite mating type. In this case, the presence of functional mitochondria in *rho⁺* cells might be a non-limiting factor for the growth rate. Hence, we assessed the number of diploid cells during the crossing experiment. We calculated the proportion of diploid cells divided by the number of haploid cells at the beginning of the experiment (Fig 3E). We found that the growth rate of diploids obtained with *rho⁺* X *rho⁰* crossings exceeded that from the *rho⁺* X *rho⁻* crossings. This result shows that the presence of *rho⁻* mtDNA in the diploid cells inhibits their growth.

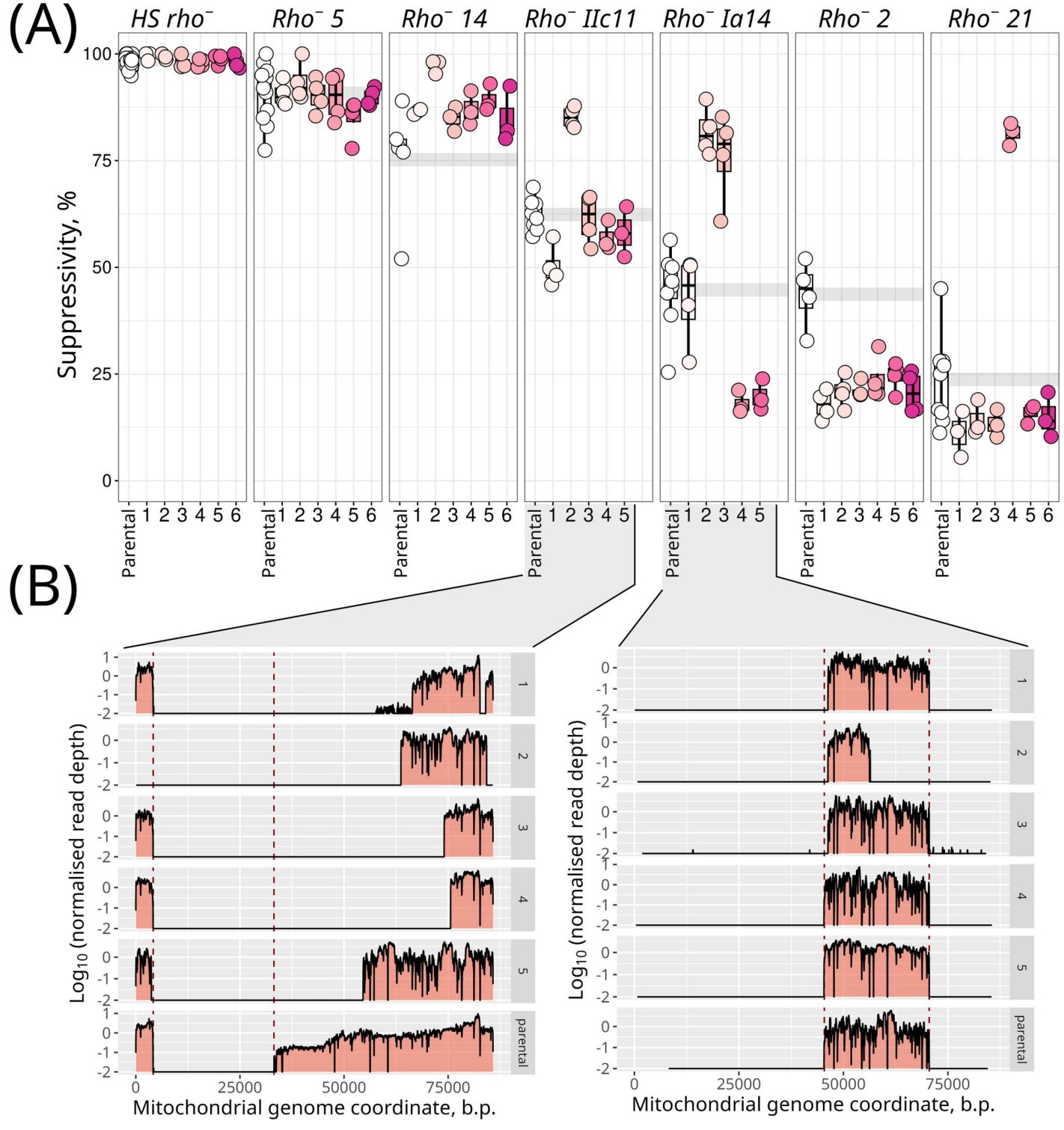

**Fig 2. Genetic bottlenecks change the suppressivity and enable the fixation of clones with secondary deletions.** (A) Suppressivity was measured in parental *rho⁻* yeast strains and their subclones derived from each strain after a narrow genetic bottleneck (single colonies). (B) The read depth of the mitochondrial DNA of the parental strains and their subclones reveals secondary deletions in some of the subclones. The read depth was normalised to the average coverage of nuclear DNA (nDNA) in the same sample and log-transformed to highlight low-coverage regions.

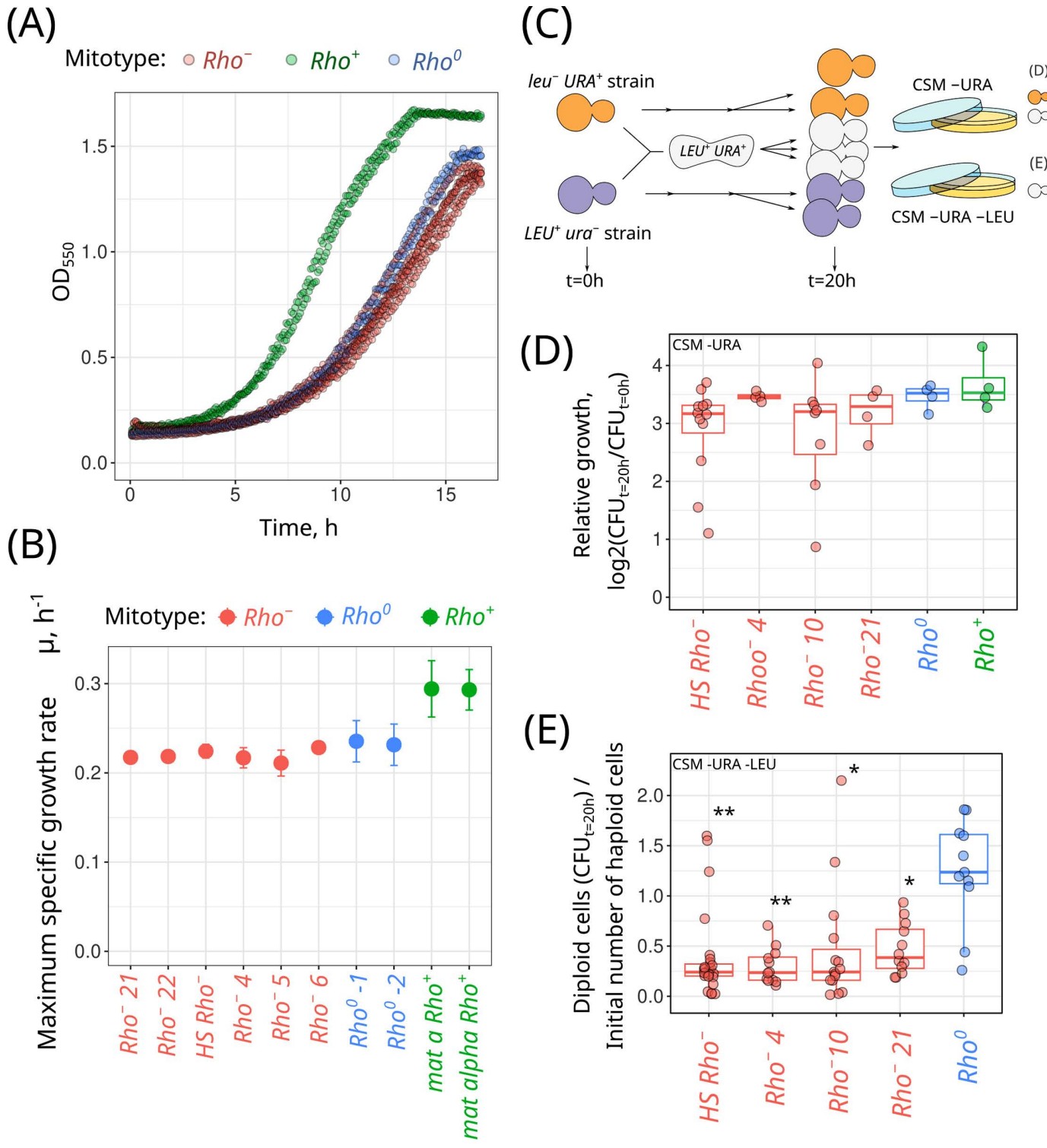

**Fig 3. Phenotypes of *rho*+, *rho*− and *rho*0 strains during the suppressivity assay experiments.** Representative growth rate (A) and maximum specific growth rates (B) of *rho*+, *rho*− and *rho*0 strains (shown as mean ± standard deviation based on the data from four independent experiments); (C) Experimental setup to measure growth rates of strains during genetic crosses; (D) Relative growth rates of yeast strains specified on the plot (*URA*+ *leu*−) crossed with *rho*+ (*ura*− *LEU*+) strain; (E) *URA*+ *LEU*+ diploid cell count produced by crossing indicated strain with (*ura*− *LEU*+) *rho*+ strain, normalised to initial cell count; *P < 0.01, **P < 0.001 (Wilcoxon rank sum exact test) comparing *rho*− X *rho*+ to the *rho*0 X *rho*+ crosses.

## Stochastic model suggests an intracellular fitness advantage of *rho*⁻ mtDNAs with deletions

Our experiments yielded estimates for three parameters of *rho*⁻ cells: suppressivity, their mtDNA copy number per cell, and growth rates in crossing experiments. We reasoned that a strain's suppressivity could be primarily derived from its growth rate and mtDNA copy number. However, it can also be influenced by the intracellular fitness of the *rho*⁻ mtDNA variant if this fitness value deviates significantly from one. In order to explore the association between these parameters, we developed a stochastic model that simulates the genetic drift and selection of mtDNAs in the heteroplasmic cells. This model takes as an input the following parameters: (1) *relative growth rates* of *rho*⁺ and *rho*⁻ strains; (2) the initial proportion of two mtDNA variants (*heteroplasmy level*), with the range of possible values taken from mtDNA copy number estimations; (3) *intracellular fitness* (imputed from the value in a range from 0 to 4). In our model, intracellular fitness was defined as the relative increase in frequency of the *rho*⁻ mtDNA variant compared to the *rho*⁺ mtDNA variant within a single heteroplasmic cell during one cell division; The model simulates random drift and selection on intracellular and intercellular levels and returns the distribution of mtDNAs in the population of cells following several cellular divisions (see S7 Fig for the in detail model description). At the end of the simulation, the model yields *suppressivity* as the ratio of cells devoid of wild type mtDNA to the total number of cells. We ran these simulations to estimate the suppressivity with varying starting parameters of the initial heteroplasmy level and intracellular fitness of *rho*⁻ mtDNA (Fig 4A). Intracellular fitness values within the range of 2–4 produced similar results for most initial heteroplasmy levels. Therefore, for further analysis, we only used the data obtained from simulations with intracellular fitness values ranging from 0 to 2. Given that intracellular fitness was an unknown variable in our simulations, we replotted the data using alternative coordinates (Fig 4B). In this representation, suppressivity is plotted against the heteroplasmy level, with the model-imputed intracellular fitness shown by a colour gradient. We subsequently overlaid our experimental data points for all 22 *rho*⁻ strains onto this plot.

Fig 4B shows that most of our experimental data points fall above the green line representing a neutral assumption that the intracellular fitness values of the *rho*⁻ and *rho*⁺ mtDNAs are equal, indicating that most *rho*⁻ mtDNAs have an

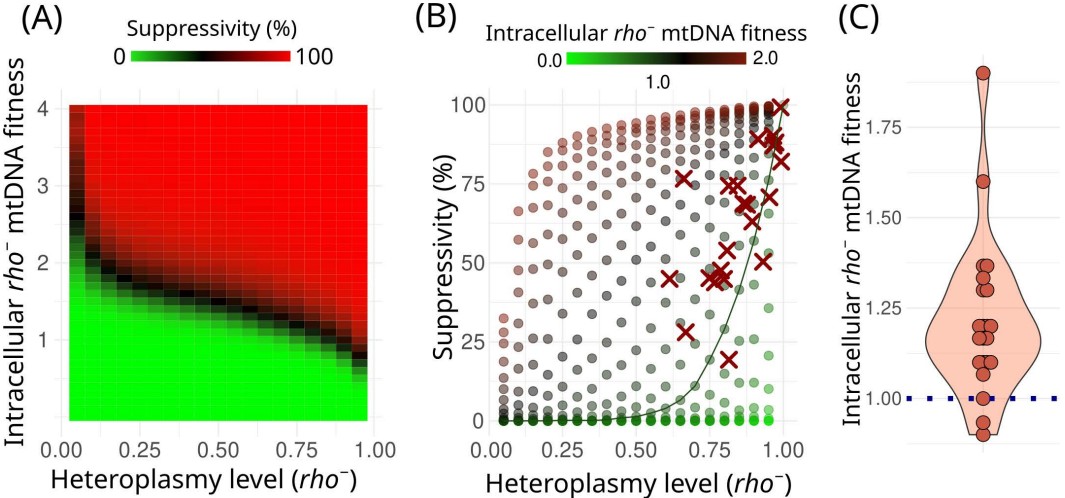

**Fig 4. Stochastic modelling suggests that the starting proportion of *rho*⁻ and *rho*⁺ mtDNA molecules in the zygotes cannot explain the observed suppressivity values.** (A) *in silico* simulation provides a suppressivity estimate for varying intracellular fitness and starting heteroplasmy level parameters; (B) Simulation data and experimental data plotted in the coordinates of starting heteroplasmy level ~ suppressivity; Points represent simulated data, crosses represent experimental data points (suppressivity and relative copy numbers of wild-type *rho*⁺ and *rho*⁻ mtDNA) for the 22 *rho*⁻ strains. A line connects the points obtained with the simulation with no replication advantage of *rho*⁻ mtDNA (Intracellular *rho*⁻ mtDNA fitness equal to 1.0); (C) Relative intracellular fitness of *rho*⁻ mtDNA variants calculated from the k-nearest neighbours in (B). P = 0.00006 for the hypothesis that the population distribution mean is equal to 1.0 according to Wilcoxon signed rank test with continuity correction.

intracellular fitness advantage compared to parental *rho*⁺ mtDNA. Furthermore, we estimated the intracellular fitness levels for our set of 22 *rho*⁻ strains from the simulation data. To do this, we determined k-nearest neighbours for each strain in the coordinates (heteroplasmy level ~ suppressivity) (k = 6) and took their average intracellular *rho*⁻ mtDNA fitness as the estimate for our strains. Fig 4C shows the distribution of the calculated fitness values and suggests that on average, the value is significantly higher than one (fitness equal to one is the neutral expectation, assuming that there is no intracellular fitness advantage of *rho*⁻ mtDNA variants).

In our model, we incorporated several assumptions. First, we posited that mutant *rho*⁻ mtDNAs have a threshold effect on yeast growth rate. Specifically, we proposed that yeast cells switch from slow to high growth when the proportion of *rho*⁺ mtDNA exceeds 0.5. This assumption aligns with the generally accepted notion that deleterious effects of mtDNAs, including *rho*⁻, are manifested when the proportion of mutants is above a certain threshold [30]. Second, we imputed into the model the copy number per cell equal to 20 which was concordant with our estimates (Fig 1C) and the published data [27]. To test the robustness of our estimates, we varied pathogenicity thresholds for *rho*⁻ mtDNAs (S8 Fig) and mtDNA copy number per cell at the moment of cell division (S9 Fig) in the simulations. Using the same approach mentioned above, we calculated the intracellular fitness advantage of *rho*⁻ mtDNA variants using simulations with different mtDNA copy numbers. The mean value of *rho*⁻ mtDNA intracellular fitness remained above one (S10 Fig), suggesting that there is an advantage within the cell for mutant mtDNAs compared to parental wild-type mtDNA, and this result is robust. Overall, our experiments and the model show that on average spontaneous deletion in yeast mtDNA provides it with a selection advantage within the cell.

## Discussion

Uniparental inheritance of mtDNA in most metazoan species prevents mtDNA recombination and imposes a problem of deleterious mutation accumulation in the mitogenomes [31]. Severe mtDNA copy number bottlenecks in the germline slow down this process [32]. In contrast to metazoa, fungi usually inherit mtDNA from both parents [33,34]. In addition, fungal mtDNAs experience frequent recombination in experimental conditions [35] and natural yeast populations [36]. Biparental inheritance enables recombination, which is likely to diminish the problem of mutation accumulation [32]. However, this makes the fungi prone to selfish mitochondrial genetic elements, which show preferential transmission to the progeny despite their deleterious effects on the whole organism. The simplest premise for the transmission bias of mtDNA variants is an increase in its copy number in one of the haploid cells — gametes. This would resemble the anisogamy typical for metazoa, where female gametes contain more mitochondria and mtDNA compared to male gametes [37].

To unravel how the mtDNA copy number in wild-type and *rho*⁻ *S. cerevisiae* cells affects their inheritance bias, we studied several *rho*⁻ strains harbouring large deletions in their mitochondrial genomes. mtDNA copy number per cell in all studied *rho*⁻ haploid strains exceeded that of the parental *rho*⁺ strain (Fig 1C). This observation is concordant with a previous study showing that *rho*⁻ yeast cells accumulated mtDNA to a copy number ~80 times higher than that in a wild-type strain [24]. Furthermore, we have recently found that the total amount of mtDNA is higher in *rho*⁻ cells, whereas in some strains, the average mtDNA amount exceeded that of *rho*⁺ cells several fold [38]. We speculate that a negative feedback loop mechanism induces mtDNA biogenesis to compensate for a deficiency in mitochondrial function that eventually leads to excessive accumulation of mtDNA. A similar mechanism was described in nematodes, where the mitochondrial unfolded protein response transcription factor ATFS-1 helped to maintain high levels of mtDNA variant harbouring a deletion by enhancing the binding of mtDNA polymerase to mutated mtDNA molecules [39]. To the best of our knowledge, such a mechanism is still unknown in yeast. Meanwhile, deletion of the *COX5* gene, encoding an essential component of cytochrome oxidase, the terminal part of the respiratory chain, increases mtDNA copy number per cell [40]. Together, these observations suggest that the ability to encode essential components of OxPhos is necessary for maintaining constant mtDNA levels across generations. If this is true, *rho*⁻ cells lacking such regulation would experience drift in mtDNA copy number, resulting in highly heterogeneous mtDNA copy numbers both per cell and per clone. This hypothesis is consistent

with the previous observation that *rho*⁻ cell suspensions are more heterogeneous in terms of mtDNA levels per cell compared to their parental *rho*⁺ strains [38]. Moreover, this hypothesis is supported by the observed variability in mtDNA copy number among *rho*⁻ strains in this study (Fig 1C–E). However, further studies are necessary to elucidate the properties and mechanisms underlying the regulatory processes linking OxPhos and mtDNA maintenance.

We did not find a positive correlation between genome size and either mtDNA copy number or suppressivity in the *rho*⁻ strains (Figs 1D and S4). On the one hand, the absence of such correlation could be explained by the absence of contribution of the sequence length to the *rho*⁻ mtDNA variant suppressivity. On the other hand, it may also result from the presence of mtDNAs with secondary deletions in the *rho*⁻ strains. Such mtDNAs if accumulated in high copy numbers can strongly affect suppressivity.

The simplest explanation of the suppressivity phenomenon could be the observed high copy number of mtDNA molecules in the *rho*⁻ strains followed by random segregation of the molecules (Fig 1C). However, by imputing mtDNA/nDNA ratios and relative growth rates of *rho*⁺ and *rho*⁻ strains we were able to explain the *rho*⁻ strains' variance in suppressivity only partially (Fig 4B). It should be mentioned that in yeast, mtDNA molecules are usually concatenated [41,42]. Meanwhile, both methods (qPCR and NGS) used in our study to assess mtDNA copy numbers cannot distinguish molecules containing one repeat unit from concatenated molecules. To estimate the intracellular fitness of *rho*⁻ mtDNA variants, we did not take into account that *rho*⁻ mtDNAs consist of a smaller number of segregating units. However, if this were the case, we would overestimate the ratio of *rho*⁻ to *rho*⁺ mtDNA in zygotes (the heteroplasmy parameter of the model) and underestimate intracellular fitness in our prediction. In other words, even if each *rho*⁻ mitotype was represented by a separate molecule segregating independently during cell division, we could not explain the observed values of suppressivity by the initial ratio of mtDNA molecules in *rho*⁻ and *rho*⁺ strains crossing. Therefore, we conclude that in yeast, some mtDNA variants with large deletions can have pronouncedly higher intracellular fitness than the full-size wild type mtDNAs.

In line with our results indicating that some *rho*⁻ mtDNAs can have higher intracellular fitness than others, we have shown that the suppressivity of yeast *rho*⁻ clones can be changed after the genetic bottleneck (Fig 2) and, on average, increase with generations (S6 Fig). We showed that passing a *rho*⁻ strain through a single colony fixes secondary mutations in its mtDNA that can contribute to the increase in its suppressivity (Fig 2B). The gradual increase in suppressivity could be attributed to two factors. On the one hand, cells with secondary mutations in mtDNA leading to a higher suppressivity proliferate faster than parental cells with the original mutation. However, Fig 3 shows that there is no pronounced difference between the growth rate of *rho*⁻ strains with different deletions. On the other hand, spontaneous secondary deletions in *rho*⁻ mtDNA could lead to the appearance of mtDNA variants with intracellular fitness exceeding the intracellular fitness of the original *rho*⁻ mitotype. Several experiments in our study support the second possibility. First, different *rho*⁻ mtDNA molecules can have various intracellular fitness (Fig 4C). Therefore, *rho*⁻ cells will eventually produce new *rho*⁻ variants with increased intracellular fitness, as evidenced by the post hoc analysis in S6 Fig. Due to an increase in intracellular fitness, this variant is expected to displace parental *rho*⁻ variants within the same cell and all descendants of this cell. In the presence of *rho*⁺ cells, such variants are expected to be removed from the population because of the selection on the cell level (see a hypothetical scheme in Fig 5A). Meanwhile, in *petite* cell populations, newly emerging mtDNA variants with low intracellular fitness are expected to be outcompeted and removed from the population, while the mtDNA with high intracellular fitness will repeatedly emerge and displace the parental variants within cell lineages (Fig 5B). We propose that as a result, a ratchet-like mechanism leads to a gradual increase in the proportion of cells carrying suppressive *rho*⁻ mtDNA within small yeast populations. However, this process should be limited by the maximal possible intracellular fitness of the *rho*⁻ mtDNA.

What could be the mechanism ensuring *rho*⁻ mtDNA intracellular fitness advantage in heteroplasmic cells? First, the advantage may be attributed to the reduced replication time of mtDNA molecules harbouring large deletions but retaining replication origins. Similar to PCR reactions, the amplification of shorter mtDNA molecules is more robust and proceeds more efficiently, especially under conditions of nucleotide deficiency. Indeed, the probability of *rho*⁺ mtDNA displacement

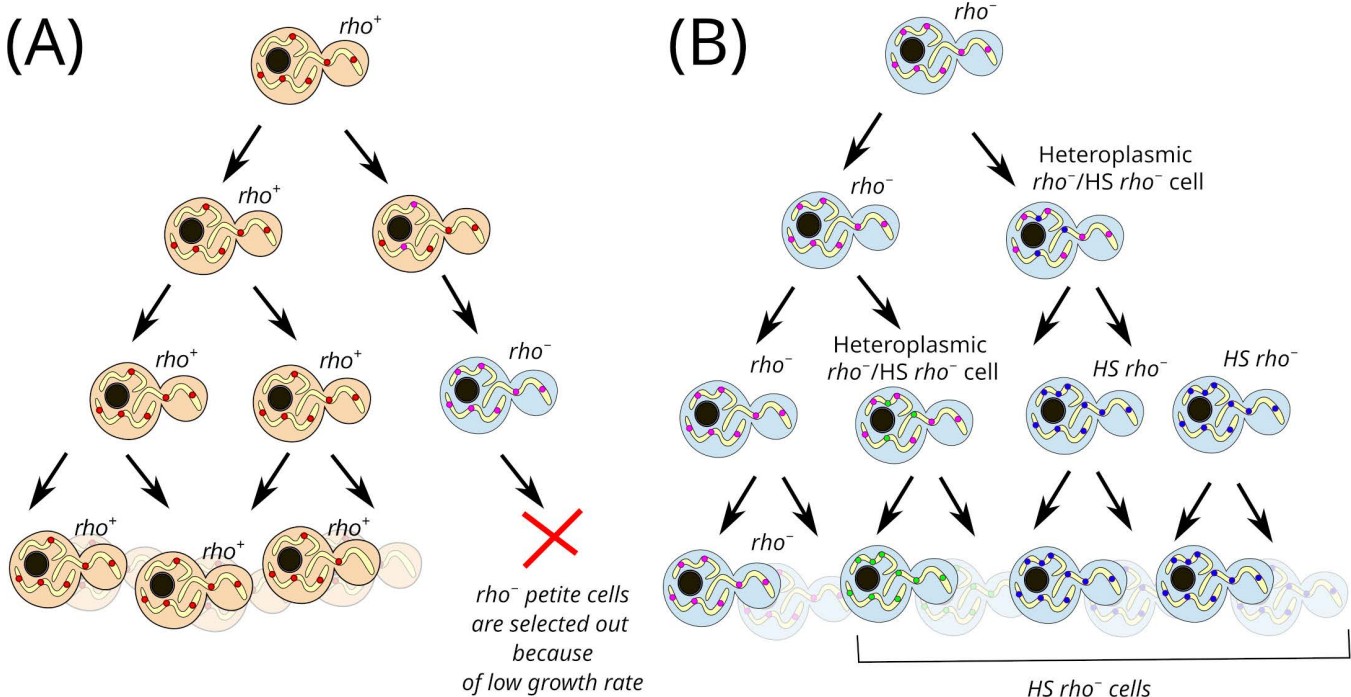

**Fig 5. MtDNA dynamics in yeast suspension, a hypothetical scheme.** (A) *rho*⁻ petite cells are continuously generated and selected out from the population of *rho*⁺ cells; (B) Intracellular selection favours the fixation of various hypersuppressive (*HS*) *rho*⁻ variants within the *rho*⁻ petite cell population.

appears to be negatively correlated with the size of the remaining mtDNA sequence [17] and positively correlated with the density of replication origins retained in a deletion variant of mtDNA [42]. Furthermore, the partial inhibition of dNTP synthesis by the overexpression of *SML1*, encoding the inhibitor of ribonucleotide reductase *RNR1*, strongly increased, and its deletion decreased the suppressivity [43]. Therefore, it might be that *rho*⁻ mtDNAs simultaneously cause problems with the biogenesis of essential DNA replication components (e.g., nucleotides and ATP) and gain a replication advantage because of these deficiencies.

Second, mtDNA may gain an advantage via preferential diffusion of mutant mtDNAs to the daughter cells. Upon cell division, yeast daughter cells receive a limited number of mtDNAs amplified from a few randomly selected clones. This leads to a rapid loss of heteroplasmy in yeast cells because the progeny receives only one or another mitotype [44,45]. This mechanism is likely to rely on the limited diffusion of mtDNA molecules and mtDNA-encoded products along the mitochondrial network. The limited diffusion creates a linkage between mtDNA genotypes and phenotypes, which is considered to be essential for quality control of mtDNAs [46,47]. Cristae appear to be an essential factor preventing the diffusion of protein components of large protein complexes along mitochondria [48]. Accordingly, deletion of MIC (mitochondrial cristae organisation) genes prevents preferential inheritance of the wild type mtDNA to a bud [49]. We speculate that *rho*⁻ mtDNAs showing intracellular advantage can avoid this sequestration, e.g., by the structural features of the spatial structure within the nucleoid. As a result, in contrast to the wild type mitotypes, all daughter cells can receive a portion of mutant mtDNAs. Whatever the mechanism, our study suggests that the inheritance bias of *rho*⁻ mtDNAs is affected by variation in the intracellular fitness of the mutant mitotypes.

Biparental mtDNA inheritance repeatedly produces mitochondrial heteroplasmy, which creates the conditions for selfish mitochondrial genetic elements to spread in the fungal populations. Here, we provide rough estimates of the intracellular fitness ratio of mtDNA variants with deletions. Our data demonstrate that yeast mtDNA variants with large deletions can

exhibit high intracellular fitness, challenging the fungal cells' ability to maintain a functional mitochondrial genome containing all essential components for translation and oxidative phosphorylation.

## Methods

### Yeast strains, growth conditions and reagents

All strains used in this study are descendants from *W303* (*ade2–101 his3–11 trp1–1 ura3–52 can1–100 leu2–3, 112, GAL, psi+*) *mat a* or *mat alpha* strains used in our previous study [25]. Rich and complete synthetic media compositions were prepared according to Sherman [50] and manufacturer's instructions (Sigma-Aldrich). All incubations were performed at 30°C with shaking (200 rpm) if not indicated otherwise.

To obtain the spontaneous *rho⁻* strains, the parental *rho⁺* strain W303 *URA⁺ mat alpha* pre-grown in liquid YPD to the exponential growth phase was plated onto YPDGly medium (containing 0.1% glucose and 2% glycerol). Colonies displaying a *petite* phenotype were amplified on solid YPD and used to assess suppressivity. Subsequent suppressivity measurements were performed from frozen culture stocks. For selected *rho⁻* strains, we also assessed suppressivity of individual subclones isolated from the stock culture by picking single colonies.

The *rho⁰* strain was obtained by incubating exponentially growing *rho⁺* cells in YPD with ethidium bromide added to a final concentration of 0.1 mg/mL. The cells were further incubated overnight (~16 hours) at 30°C with shaking, then diluted and plated on solid YPDGly medium (0.1% D-glucose and 2% glycerol) to obtain approximately 200 single colonies. Selected colonies with a *petite* phenotype were confirmed to be *rho⁰* by their inability to grow on glycerol-containing media (YPGly) and the absence of DAPI-positive foci in the cytoplasm (see S11 Fig). For this, cells were fixed in 70% ethanol and incubated with DAPI at a final concentration of 2 µg/mL for 30 minutes, then resuspended in 1x PBS and analyzed under Olympus BX51 microscope with the following filter set: U-MNU2 filter for DAPI (excitation $\lambda = 360\,nm - 370\,nm$; dichroic mirror $\lambda = 400\,nm$; emission wavelength $> 420\,nm$).

### Yeast DNA isolation and quantitative PCR

To measure mtDNA/nDNA ratio, we took exponentially growing yeast cells and isolated total DNA using a standard protocol which is based on the preparation of protoplasts and their lysis using SDS [51]. We carried out qPCR on the CFX96 Touch Real-Time PCR Detection System according to the manufacturer's instructions. To quantify PCR products, we used EvaGreen-ROX (Syntol) as the dsDNA probe. To detect mtDNA copy numbers in *rho⁻* mutants, we designed three pairs of primers annealing to the different loci of the mtDNA genome (S3 Table). As a reference to the nuclear DNA we used primers to the *ACT1* gene (S3 Table).

### Library preparation and next generation sequencing of mitochondrial genomes

*HS rho⁻* strain was sequenced using Illumina MiSeq platform and MiSeq Reagent Kit Version 3 (150 cycle). Selected subclones of two *rho⁻* strains were sequenced with the Illumina NovaSeq6000 platform (100 cycle, paired-end sequencing). All other strains were sequenced using Illumina HiSeq platform, using HiSeq Flow Cell v3, TruSeq SBS Kit v3 (50 cycle) and TruSeq PE Cluster Kit v3 and on the Illumina Nextseq platform, using the Nextseq Mid Output Kit (150 cycle, paired-end sequencing). Libraries were prepared using the TruSeq RNA Library Prep Kit or NEBNext Ultra II DNA Library Prep Kit for Illumina and the NEBNext Multiplex Oligos for Illumina (Index Primers Set 1).

Obtained reads were trimmed with trimmomatic v. 0.32 with default parameters [52] and mapped with bowtie2 [53] to the reference S288C *Saccharomyces cerevisiae* genome (sacCer3, NCBI RefSeq assembly GCF_000146045.2).

### Quantification of mtDNA to nDNA ratio using Next Generation Sequencing data

To quantify mtDNA copy numbers, we compared read coverage of the mitochondrial and nuclear genomes; this approach has been previously used in other studies [26,54]. First, we mapped the sequence reads to the reference *S. cerevisiae*

genome using bowtie2 [53]. After this, we calculated the nDNA average read coverage using Rsamtools ([https://bioconductor.org/packages/Rsamtool](https://bioconductor.org/packages/Rsamtool)). To negate the effect of rRNA repeat and telomeric regions which have higher read counts, we discarded nuclear genomic regions with extreme quantiles from the analysis. Thereafter, we calculated average mitochondrial genome read coverage. Given poor coverage of AT-rich genomic regions, to quantify mitochondrial genome read coverage, we analysed the coverage only in the GC-rich regions of the mitochondrial genome. Next, we calculated the coverage in the positions used for the quantitative PCR (see S1 Fig). Given that *rho⁻* strains retained random regions of the parental genome, we calculated the average coverage in all three regions separately and then took the maximum of the obtained values.

Relative read depth in different regions of mtDNA is presented in S2 Table.

## Suppressivity assessment

Suppressivity was estimated as the proportion of *petite* diploid colonies in the total number of diploid progeny as described previously [55] with several modifications. Prior to the experiment, strains of two different mating types were grown in rich medium YPD to exponential phase ($OD_{550}$ = 0.4-0.8). Strains were mixed in a 2 mL eppendorf tube or 96-well plate to the final $OD_{550}$ = 0.1 of each strain ($2 \times 10^6$ cells/mL) in the total volume of 200 µL. Matings were performed at room temperature without mixing. After 20 h of mating cell suspensions were diluted with mQ water (10–1000 times depending on the mating efficiency for the given pair of strains) and plated on double-selective media (e.g., YNB -LEU -URA) containing 2% glycerol and 0.1% glucose as carbon sources. Such media allows selecting only diploid cells carrying both selective markers and furthermore distinguishing *petite* and *grande* diploid colonies. After 2–3 days we counted the number of *petite* and *grande* colonies and calculated the suppressivity.

## Assessment of ethidium-bromide-induced mtDNA loss

To quantify the proportion of *rho⁻* cells retaining mtDNA after incubation with ethidium bromide, cells were pre-grown to $OD_{550}$ = 0.1 in YPD, then incubated in 2 mL YPD with 25 µg/mL ethidium bromide for 16 hours. After that, cells were washed once with YPD and incubated in fresh YPD for 6 more hours. To check for mtDNA retention, cells were stained with DAPI as described above. 300–1000 cells were counted per strain for the presence of the DAPI signal in the cytoplasm.

## Growth rate assessment

Growth rate measurement was performed in a spectrophotometer (SpectrostarNANO) with automated $OD_{550}$ measurement at 5-minute intervals. Analysed strains were pre-grown in YPD to exponential phase, then diluted in YPD to an equal density of $10^6$ cells/mL ($OD_{550}$ = 0.05) and inoculated into a 96-well plate (Eppendorf) at 100 µL per well. Plates were incubated in the spectrophotometer at 30°C for up to 18 hours, with orbital shaking before measurements (500 rpm for 30 sec).

To determine the maximum growth rates, we ln-transformed the OD values and computed the slopes for each sliding window (50 points in size). The maximum value was selected; see the script and example fitting available on GitHub: [https://github.com/Knorre-lab/GrowthCurves](https://github.com/Knorre-lab/GrowthCurves)). It should be noted that the calculated maximal growth rate ($\mu_{max}$) values may be biased, as they were measured in batch culture experiments where OD may not linearly depend on cell density in dense cultures.

To assess growth rates of *rho⁺* and *rho⁻* strains in the conditions of the suppressivity-measurement experiment, we quantified CFUs in the spot assays sampled from the crossing cultures. Crossings were set as described above, by mixing two strains to a final $OD_{550}$ = 0.1 of each strain ($2 \times 10^6$ cells/mL) in a total volume of 200 µL, and incubated at room temperature for 20 h without mixing. At the first (0 h) and last (20 h) timepoints, 10 µL of the mating cultures were used to prepare five serial dilutions in sterile mQ water (1:10–1:10⁻⁶ depending on the mating efficiency). Each dilution was plated on YPD, single-selective (YNB -LEU and YNB -URA) and double-selective (YNB -LEU -URA) media containing 2%

glucose. CFUs were quantified after 1 (for YPD) or 2 days (for selective media) of incubation at 30°C. Only diploid progeny could form colonies on the double-selective medium, while the single selective medium allowed growth of only one of the parental strains, and YPD allowed growth of all cells. These numbers were used to calculate the growth rate of both diploid progeny and either of the parental strains.

## Stochastic model

To simulate genetic drift and dual level of selection in yeast we wrote a stochastic model using Julia language. The script is available here: (https://github.com/dknorre/YeastHeteroplasmy and illustrated in S7 Fig. An individual cell was defined as an entity which can harbor a predefined number of wild type and mutated mtDNA molecules, has an individual *grande* or *petite* phenotype tag and the parameter indicating the time spent after the previous cell division. During each simulation, the total number of mtDNA molecules within a cell remains constant, while the proportion of mutated and wild-type mtDNA can change due to selection and stochastic drift.

In each simulation, (1) a set of $10^3$ cells with specified levels of $rho^-$ and $rho^+$ molecules is generated. The time elapsed since the last cell division is assigned randomly to each cell, which helps to avoid potential artifacts that might arise from the simultaneous division of cells, specifically the synchronisation that occurs at the start of the simulation. (2) For each cell, the algorithm calculated the phenotype which was set as a function of $rho^-/rho^+$ mtDNA molecules ratio. In this study, we assumed the threshold dependencies of the phenotype on the ratio of $rho^+/rho^-$ mtDNAs, although to improve robustness, we tested different values of the threshold (S8 Fig). The *petite* and *grande* phenotypes were represented by duplication time, i.e., the number of minutes necessary to produce a new daughter cell. The values were taken from the experiments in Fig 3B and set as 140 minutes for *grande* phenotype and 210 minutes for *petite* phenotype. However, it should be noted that the actual growth rates of *grande* and *petite* cells in the diploid state during crossing experiments may differ from these values. (3) The simulation incremented time which relates to cell division. Each cell that reached its duplication time (according to the phenotype) divided and produced two cells with randomly distributed $rho^-$ and $rho^+$ mtDNAs. To do this, we randomly sampled $rho^+$ and $rho^-$ mtDNAs into the new cells. Importantly, before the sampling, we doubled the number of $rho^+$ mtDNAs and multiplied the number of $rho^-$ mtDNAs by 2 × Intracellular fitness value. (4) After this, we returned to step two and repeated the simulation for 1440 minutes.

## Data analysis and visualisation

All data analysis and generation of figures were performed using R language version 4.4.0 and tidyverse libraries [56]. Where appropriate, we have displayed individual data points alongside boxplots using default parameters from the R ggplot2 library. These individual data points represent either separate strains or measurements taken on different dates, depending on the experimental setup. By default, the boxplot borders denote the first and third quartiles, while the whiskers extend from these hinges to the largest and smallest values within 1.5 times the interquartile range (IQR). The values outside of this range are plotted as outliers. Circular map with mtDNA deletion variants were generated with circlize R package [57].

## Supporting information

**S1 Table. Deletion coordinates of *rho*− strains.** The coordinates were deduced from short-read sequencing assemblies and, for some strains, verified by PCR and Sanger sequencing.
(DOCX)

**S2 Table. Quantification of mtDNA copy number from NGS data: depth of reads mapped to mtDNA normalised to depth of reads mapped to the nuclear genome.** Trim mean and median values were calculated for the entire mitochondrial genomes. In the rest of the columns, the average read depth was calculated in specified positions.
(DOCX)

**S3 Table. Primers used for quantitative PCR (qPCR) of mtDNA to nDNA ratio.**
(DOCX)

**S4 Table. Quantification of mtDNA copy number using qPCR.** The mtDNA copy number was determined as the ratio of mitochondrial to nuclear genome loci abundance, calculated as -logCt. The positions of the amplified regions 1 and 3 used for the mtDNA quantification are illustrated in S1 Fig.
(DOCX)

**S5 Table. Raw data of growth characteristics for yeast haploid and diploid strains used in Fig 3.**
(XLS)

**S1 Fig. Relative position of the regions used for mtDNA quantitative PCR and the regions retained in *rho*− mitochondrial genomes.**
(PDF)

**S2 Fig. Concordance between mean qPCR estimates and NGS-based estimates of mtDNA copy numbers.** Kendall's rank correlation tau = 0.683, p-value = 8.266e-05.
(PDF)

**S3 Fig. Correlation of *rho*− mtDNA copy number per nuclear genome estimated from NGS data with suppressivity (as in Fig 1C, but with the strain names added as point labels).**
(PDF)

**S4 Fig. No apparent correlation between the genome lengths of *rho*− mtDNA and their suppressivity.**
(PDF)

**S5 Fig. The ability of *rho*− yeast cells to retain mtDNA upon growth with the DNA-intercalating agent ethidium bromide correlates with the *rho*− strain's suppressivity (Kendall's rank correlation tau = 0.56, p-value = $8×10^{-4}$).**
(PDF)

**S6 Fig. Suppressivity increases within yeast subclones.** (A) Schematic showing the experimental setup for measuring the suppressivity drift over generations; (B) Change in suppressivity between the first (green circles) and subsequent (orange and blue circles) assessments (n = 76). The orange vs blue color of the circle illustrates the direction of change in subsequent experiments compared to the first one, with orange indicating a decrease and blue indicating an increase of suppressivity over generations. X labels the results of individual experiments.
(PDF)

**S7 Fig. mtDNA selection in heteroplasmic yeast cells: a simulation guided by experimental data.** (A) Simulation algorithm, see details in the Methods section. (B) The simulation predicts the proportion of cells containing only *rho*+ mtDNA (represented by the blue area) and those with only *rho*− mtDNA (represented by the red area). Each plot represents an overlay of the results of ten simulations, to overlay blue and red areas that illustrate homoplasmic cells we set their opacity to 0.1. The not coloured areas correspond to the cell retaining heteroplasmy. Three plots represent simulation results with different intracellular fitness levels of *rho*− mtDNA (1, 1.3, and 1.6). The relative cell-level fitness of *rho*− cells was simulated using doubling times of 140 minutes and 190 minutes. The pathogenicity threshold was set at 0.5, indicating that a *rho*− cell-level phenotype only manifests when more than half of the mtDNA in a cell is *rho*−.
(PDF)

**S8 Fig. Variation of pathogenicity threshold in the simulations of suppressivity.** Simulation data and experimental data plotted in the coordinates of starting heteroplasmy level ~ suppressivity (same as in the Fig 4B); Points

represent simulated data, crosses represent experimental data points for the 22 *rho⁻* strains; a line connects the points obtained with the simulation with no replication advantage of *rho⁻* mtDNA (Intracellular *rho⁻* mtDNA fitness equal to 1.0).
(PDF)

**S9 Fig. Variation of mtDNA copy number (nmtDNA) in the simulations of suppressivity.** Simulation data and experimental data plotted in the coordinates of starting heteroplasmy level∼suppressivity (same as in the Fig 4B); Points represent simulated data, crosses represent experimental data points for the 22 *rho⁻* strains; a line connects the points obtained with the simulation with no replication advantage of *rho⁻* mtDNA (Intracellular *rho⁻* mtDNA fitness equal to 1.0).
(PDF)

**S10 Fig. Relative intracellular fitness of *rho−* mtDNA variants calculated from the k-nearest neighbours for simulations with different mtDNA copy numbers (number of mtDNA segregating units).** Individual lines illustrate the predictions for individual *rho⁻* strains, and the red bold line shows the average value.
(PDF)

**S11 Fig. DAPI staining of *rho*0 and *rho−* cells.** Arrows indicate putative mitochondrial nucleoid foci in *rho⁻* cells. Images are representative and were acquired and processed independently on different days.
(PDF)

## Acknowledgments

We are grateful to Maria Kasyanova for the valuable help in confirming deletion sites in mtDNAs and Ekaterina Smirnova for conducting some of the crossing experiments. We acknowledge the Skoltech Genomics Core Facility for their assistance in sequencing the yeast subclones.

## Author contributions

**Conceptualization:** Nataliia D. Kashko, Dmitry A. Knorre.

**Data curation:** Nataliia D. Kashko, Dmitry A. Knorre.

**Formal analysis:** Nataliia D. Kashko, Sofya K. Garushyants, Dmitry A. Knorre.

**Funding acquisition:** Dmitry A. Knorre.

**Investigation:** Nataliia D. Kashko, Georgii Muravyov, Iuliia Karavaeva, Dmitry A. Knorre.

**Methodology:** Nataliia D. Kashko, Georgii Muravyov, Iuliia Karavaeva, Elena S. Glagoleva.

**Project administration:** Dmitry A. Knorre.

**Resources:** Nataliia D. Kashko, Elena S. Glagoleva, Maria D. Logacheva, Dmitry A. Knorre.

**Software:** Sofya K. Garushyants, Dmitry A. Knorre.

**Supervision:** Dmitry A. Knorre.

**Validation:** Nataliia D. Kashko, Sofya K. Garushyants, Dmitry A. Knorre.

**Visualization:** Nataliia D. Kashko, Dmitry A. Knorre.

**Writing – original draft:** Nataliia D. Kashko, Dmitry A. Knorre.

**Writing – review & editing:** Nataliia D. Kashko, Georgii Muravyov, Iuliia Karavaeva, Elena S. Glagoleva, Maria D. Logacheva, Sofya K. Garushyants, Dmitry A. Knorre.

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
