## [Decision Letter · Decision Letter 0]

Feb 21 2025

PGENETICS-D-24-01121

Inheritance bias of deletion-harbouring mtDNA in yeast: the role of copy number and intracellular selection

PLOS Genetics

Dear Dr. Knorre,

Thank you for submitting your manuscript to PLOS Genetics. After careful consideration, we feel that it has merit but does not fully meet PLOS Genetics's publication criteria as it currently stands. Therefore, we invite you to submit a revised version of the manuscript that addresses the points raised during the review process. The revised manuscript will then be fully evaluated.

Please submit your revised manuscript within 60 days Feb 21 2025 11:59PM. If you will need more time than this to complete your revisions, please reply to this message or contact the journal office at plosgenetics@plos.org. Please include the following items when submitting your revised manuscript:

We look forward to receiving your revised manuscript.

Kind regards,

Ashok Bhagwat, Ph.D.

Academic Editor

PLOS Genetics

Geraldine Butler

Section Editor

PLOS Genetics

Aimée Dudley

Editor-in-Chief

PLOS Genetics

Anne Goriely

Editor-in-Chief

PLOS Genetics

**Additional Editor Comments:**

Your manuscript was thoroughly evaluated by three experts in the field and all find it to be interesting and a potential advance in the field of yeast genetics. However, all the reviewers raised several important points that may require additional experimentation, clarification of experimental methods, and better data analysis and presentation. Some of the reviewer's comments directly challenge the conclusions of this manuscript. The technical concerns of the reviewers include the accuracy of measurement of cell density and copy number of mitochondrial DNA, and the selection of parameters for the stochastic model for cell fitness. Their comments also suggest that many experimental details are missing from the manuscript.

**Journal Requirements:**

At this stage, the following Authors/Authors require contributions: Nataliia D. Kashko, Elena S. Glagoleva, Iuliia Karavaeva, Maria D. Logacheva, Sofya K. Garushyants, and Dmitry A. Knorre. Please ensure that the full contributions of each author are acknowledged in the "Add/Edit/Remove Authors" section of our submission form.

The list of CRediT author contributions may be found here: https://journals.plos.org/plosgenetics/s/authorship#loc-author-contributions

https://journals.plos.org/plosgenetics/s/submission-guidelines#loc-parts-of-a-submission

- ® on Line: 411.

- TM on Line: 411.

5) We have noticed that you have uploaded Supporting Information files, but you have not included a list of legends. Please add a full list of legends for your Supporting Information files after the references list.

Potential Copyright Issues:

- Figure S4. Please confirm whether you drew the images / clip-art within the figure panels by hand. If you did not draw the images, please provide a link to the source of the images or icons and their license / terms of use; or written permission from the copyright holder to publish the images or icons under our CC BY 4.0 license. Alternatively, you may replace the images with open source alternatives. See these open source resources you may use to replace images / clip-art:

**Reviewers' comments:**

Reviewer's Responses to Questions

**Comments to the Authors:**

Reviewer #1: Kashko et al illuminate the interesting question of why, following the mating of gametes with mutated and intact mtDNA respectively, mutated mtDNA take over in diploid cell populations expanded from zygotes, despite diploid cells with mutated mtDNA performing worse than diploids cells with intact mtDNA. They distinguish between two not mutually exclusive explanations: first, that mutated mtDNA reach higher copy numbers in haploid gametes, and therefore become more numerous in zygotes, and second, that mutated mtDNAs have an intracellular competitive advantage relative intact mtDNAs in diploids. They collect data on mtDNA copy number for mutated and intact mtDNA in haploids, and measure their effect on cell fitness, approximated as maximum growth rate. They also measure the capacity of mutated mtDNA to replace intact mtDNA, when gametes are hybridised to generate diploid cell populations. Finally, they explore using population genetics/dynamics simulations in what scenarios simulated cells with mutated mtDNA comes to dominate in diploid cells obtained from mating gametes with intact and mutated mtDNA respectively. Based on these simulations, they conclude that both higher copy number of mutated mtDNA in haploid gametes, and their ability to outrival intact mtDNA within diploid cells, helps mutated mtDNA take over - even if mutated mtDNA reduce the cell fitness. I very much like the paper’s combination of experiments and simulations, which surely must be the right-way forward to address this interesting problem – but I do have a number of reservations.

1. The methods section is sparse, making it hard to evaluate whether experiments and simulations are soundly designed and data can be trusted.

A) A more detailed description of nutrient and energy concentrations, pH and how the pH is maintained would be welcome, given that the petite slow growth phenotype is essentially an intracellular aminoacid deficiency, that likely is influenced by external concentrations and the protonmotive force. It is also not clear to me how cultures and precultures were done or which experiments were performed on rich and synthetic medium respectively. Likewise, the genotypes of the W303 strain used is not given.

B) I am not sure how the rho0 control was obtained, or more generally what operative distinction between rho0 and rho- the authors use. Rho- cells may retain small mtDNA segments, often circularized, that NGS sequencing and qPCR fail to detect unless care is taken - making them appear like rho0. A good rho0 control would be e.g. a mip1delta strain. C) the number and type of replicates and how these are handled statistically are not clear to me, nor are error bars and whiskers explained in all figures.

D) I cannot find a methods section describing how the EtBr experiments and associated microscopy was performed.

E) I cannot find a description of the growth experiments and how maximum growth rates were extracted measured. I do have some concerns here. From Fig 3A, it appears that OD550, was measured while OD 600-610 is more normal. However, optical density and cell density is not linear for most spectrometers. This is normally accounted for by diluting cultures down a more linear part of the OD interval, but from the reported OD values, this doesn’t seem to have been done here. Unless this is accounted for (see e.g. PMID 12489126), the observed OD may be very misleading, as may be the extracted maximum growth rates. Moreover, it is not possible to tell to which extent extracted maximum growth rates because Fig 3A is shown with a linear y-axis, and hence the exponential phase, if any, is impossible to identify. F) The model is very sparsely described to the extent that I cannot follow the reasoning. As I understand it, cells are treated as individual agents, but mtDNA molecules are not, but their copy number is a property of cells? I am left in the dark about the assumptions the model is based on and the settings of parameters and variables. Aging is referred to but it is not clear whether that is replicative or chronological aging, or what distribution age follows, and it is not clear what phenotypes vary with age. The phenotypes of cells are not explicitly defined or referred to. I am left in the dark about important parameters, such as population size, mutation and recombination rates for mtDNA and nuclear genomes, mating, cell lag times, between cell variability, and any distinction between circular, linear or tandem arrangements of mtDNA. It is unclear whether all cells divide at the same time, and whether cell division times are the same for haploids and diploids. The reasoning and potential shortcomings and strengths of the choices made by the authors in their modelling are not well illuminated and I cannot evaluate which scenarios are well or poorly reflected in the model.

2. I am not convinced by how the authors extract and model intercellular fitness. First, the maximum growth rate (depending on how it is defined) may not reflect the average growth rate across the entire growth phase, and may thus not be reliable as a doubling time approximation. It e.g. looks like in Fig 3A rho+ cells have a growth rate advantage in early growth, but not in later growth phases. Moreover, it looks like (hard to tell, unless the non-linearity of OD and population size are taken into account, and data are shown on a log-scale) there are differences in lag-phase and the final cell yield, between rho+, - and 0. Presumably, these may contribute to cell fitness differences, but this is does not appear to be mentioned nor considered in models? The Methods says that cell doubling times, which seem to be the only determinants of cell fitness (although this is not made explicit), are taken from Fig 2 - but there is not information on cell growth in Fig 2. Presumably, the authors mean Fig 3B? However, it is unclear if the shown growth rates are for haploids or diploids, and whether they are applied to haploid or diploids in the simulation, or both. There are often substantial differences in growth between haploids and diploids. Finally, the relative fitness of rho- in Fig S5 simulations were set to 0.75, and it is not clear why or how this value relates to the doubling time numbers indicated in the Methods. There are ways to convert a doubling time differences into a selection coefficient, with various assumptions, but it’s not clear what has been done here.

3. The author’s estimates of the true cell fitness of mutated and intact mtDNA are unlikely to be exact (see above). Moreover, true cell fitness may vary between nuclear genotypes and environments. E.g. ref 16 highlight a situation where the effect of mtDNA on cell fitness is very different from the one modelled here. To me, it would seem very informative to know under what cell fitness assumptions mutated mtDNA, with different intracellular fitness advantages and different heteroplasmy levels, replace intact mtDNA in diploid cell populations. I would invite the authors to not treat fitness as a fixed parameter but as a variable in their simulations.

Minor comments:

4. When discussing the yeast petite phenotype, it would be informative if the authors explained the mechanistic origin of the slow growth phenotype (PMID 34799698), which could influence interpretations of data.

5. Line 138 Fig S2 does not support variation from 3.0 copies, but rather log2 = 2.5 and is probably not the right reference for this statement.

6.Line 147: I cannot find a measure of the correlation in either figure or text.

7. Line 433: I am not sure why a maximum is used? Presumably, deviations from the true copy number would be equally likely to go in both directions – and a median or average would seem more appropriate?

8. Line178: Are these strains and the method with which they were obtained described?

9. Line181: Do the authors have a rough estimate or interval for the number of generations? As is, it is hard to interpret the associated statements.

Reviewer #2: In their article titled “Inheritance bias of deletion-harbouring mtDNA in yeast: the role of copy number and intracellular selection” Kashko et al. investigate the phenomenon known as mtDNA suppressivity in which mtDNA molecules harbouring deletions (rho-) compete with or even entirely outcompete intact mtDNA. The study examines two factors contributing to suppressivity: higher mtDNA copy numbers in cells with deletions and a competitive advantage of these deletion-containing mtDNAs within cells. The study uses experimental data from 22 yeast strains and a computational model to determine that both factors play significant roles. The findings suggest that mtDNA deletions can have a selective advantage within the cell, even if detrimental to the organism. The study also explores the influence of genetic bottlenecks on mtDNA suppressivity and discusses potential mechanisms driving this intracellular selection.

The study is interesting, because it provides insight into the broader question how cells maintain the integrity of their mtDNA. The phenomenon that mutant mtDNA’s can outcompete intact mtDNA has been known for a while, and it has been discussed that this is due to a replicative advantage of rho- due to their shorter length or higher density of replication origins. This study sheds further light on the phenomenon of mtDNA suppressivity and provides an interesting stochastic model that helps to understand the mecnahisms underlying suppressivity. While interesting, I think the study would benefit from a slightly more detailed analysis of the sequence features in the rho- genomes that are responsible for the suppressivity.

Points to be addressed:

1. The sequencing of the rho- genomes is a great resource. Figure 1B is nice, but it is difficult to evaluate what features (origins of replication) may correlate with suppressiveness. Perhaps it would be better to depict this figure in a linear format and sort the sequences based on the starting point. The authors state that they 'did not find a strong association between the retained replication origins and suppressivity', a figure detailing this statement would be helpful. The authors state that 'two low-suppressive strains, namely IIa10 and 13, lack all three active replication origins: ORI2, ORI3, and ORI5'. How are such genomes replicated.

2. The high copy number of mtDNA in some rho- strains is astonishing. According to the graph in figure 1D two strains, one of which has a 50kB mtDNA, have a mtDNA copy number of approximately 2^9=512. To validate these results, but also to characterize the effects on the mitochondrial network, the authors should perform a DAPI staining of the cells and perform quantitative fluorescence microscopy.

3. For better assessment of the data, it would be helpful to include heatmaps in figures C, D and E (similar to Fig S2) that display the mtDNA genome length (for C), the suppressivity (for D) and the mtDNA genome length (for E). It would also be helpful to label the dots that that represent rho- genomes that are used in figure 2. Perhaps that could also be done in a supplementary figure.

4. It is very interesting to see in Figure 2 that the suppressivity changes in subclones. However, the molecular reasons for that remain unknown. It would be very interesting to sequence the subclones to determine, whether secondary mutations within the mtDNA, as the authors discuss, are found. I feel that such an experiment would strengthen the molecular insight of the story.

5. The stochastic model is interesting. However, I have problems understanding how the mtDNA copy number was integrated into the model. The determination of the mtDNA copy number in the rho- strains indicates greatly varying numbers. In Figure S5 it is stated that both of the parental cells contribute 10 mtDNA copies into the zygote. How are these numbers linked to the 'Heteroplasmy level (rho-)' (x-axis Figure 4A)? More information on how the simulation was set up would be helpful for the reader to interpret the findings. I am somewhat confused about the 'Intracellular rho- mtDNA fitness' value. In the text it is stated that it is within 0 and 3. In Figure 4A it is between 0 and 4 and in Figure 4B it is between 0 and 2. What is the reasoning behind this?

Other points:

1. I find Figures 3C and 3D difficult to follow. More information should be given in the text about the experiment or the figure. What strains carry the URA+-marker? How long are the cells incubated? What was the expectation?

2. What is the difference between 'Suppressiveness' and 'Suppressivity' in Fig S3? I assume it is the same, but then it seems unnecessary to include a heatmap

3. What is Ne in Figure S4?

4. Line 139: Some context should be given to why the results of 3-15-5 copies of mtDNA per cell is unrealistic.

5. Line 172: What is meant by 'convergent'?

6. Line 178: A little more context to the 76 strains would be necessary. I assume those are rho- strains, but this should be stated.

7. Lines 292-294: The authors speculate that the increase of mtDNA in rho- cells is due to a compensation mechanism. Why do the copy numbers vary so drastically between the strains. The authors could comment on this. Line 299: There are many deletions that lead to compromised OXPHOS. Is the deletion of COX5 the only one that leads to more mtDNA?

Reviewer #3: The manuscript by Kashko and collaborators argues for a model in which rho- mitotypes become dominant in heteroplasmic diploid cells due to increased copy number and preferential duplication/selection during growth. The current manuscript is quite interesting but the main arguments would be improved by some additional analyses.

A. The author’s model for the mechanism of suppression by rho- variants is that the rho- mtDNA is replicated/inherited more readily and can displace the rho+ mtDNA within a cell lineage. This model has several testable hypotheses that have not been addressed in the current manuscript but should be.

1. In a cross between a suppressive rho- strain and a rho+ strain, the rho- diploids should have mtDNA that matches the rho- strain, which can be easily determined using NGS.

2. In a cross between a suppressive rho- strain and a rho+ strain, the rho+ diploids should be heteroplasmic intermediates, and both rho- and rho+ mtDNA should be observed in the NGS.

B. The paper would be greatly strengthened by more detailed analysis of the variability of suppressivity observed after genetic bottlenecks. The authors claim (lines 173-174) that their results are suggestive of additional mtDNA mutations in rho- strains. The authors should do NGS on the subclones with large changes suppressiity (e.g. Rho- la14, Rho- 2, and Rho- 21, or the equivalent). Are their additional mutations? Or do changes in suppressivity correspond to subclones with increased/decreased rho- mtDNA copy numbers, which would be more in line with the authors’ model.

C. I think Fig 5B is misleading in the context of the data for this paper. This figure attempts to illustrate the authors’ suggestion of a “ratchet” in which more suppressive mtDNA variants will overtake a rho- cell population. However, the authors have not performed this experiment (though they could). Note that this mechanism requires that (1) suppressive mtDNA variants are constantly arising (not demonstrated in this manuscript) or (2) there is a selective advantage for suppressive mtDNA variants (which is in contrast to the authors’ data).

A much better Fig 5B that would illustrate the gradual loss of the rho+ mtDNA within a heteroplasmic population with a suppressive rho- mtDNA and would serve to explain why rho+ cells dominate in normal populations (Fig 5A) but why rho- diploids are recovered with suppressive heteroplasmic mtDNAs.

Minor comments.

How many nuclear genome mutations are recovered in the spontaneous rho- strains? Do any affect genes involved in mitochondrial maintenance and aerobic respiration?

Fig 1B would be more interpretable if the rho- genomes were sorted in some way (e.g. by suppressivity?) For example, it appears that the most suppressive mitotypes include both ORI2 and ORI7 in contrast to the authors’ assertion (lines 114-117) that there weas no association between replication origins and supressivity.

To improve comparison with previous studies that have used the mtDNA copy number from 14,000-20,000, the authors should define a correction factor in the rho+ strain (and potentially other strains) that allows NGS-based copy numbers from their three chosen regions to be transformed into a 14,000-20,000 NGS copy number measurement. Does this correct the differences in mtDNA/nDNA ratios between NGS and qPCR methods?

The axes in Fig 1D should be switched for consistency between Figs 1C-1E.

The authors should add a panel to Fig 1 displaying suppressivity as a function of genome length.

**Have all data underlying the figures and results presented in the manuscript been provided?**

Reviewer #1: **No: ** All data will need to be provided

Reviewer #2: **No: ** Not sure if the sequencing data are available

Reviewer #3: Yes

PLOS authors have the option to publish the peer review history of their article (what does this mean? ). If published, this will include your full peer review and any attached files.

**Do you want your identity to be public for this peer review?** For information about this choice, including consent withdrawal, please see our Privacy Policy .

Reviewer #1: No

Reviewer #2: No

Reviewer #3: No

**Figure resubmission:**
---

## [Decision Letter · Decision Letter 1]

Dear Dr Knorre,

We are pleased to inform you that your manuscript entitled "Inheritance bias of deletion-harbouring mtDNA in yeast: the role of copy number and intracellular selection" has been editorially accepted for publication in PLOS Genetics. Congratulations!

Yours sincerely,

Ashok Bhagwat, Ph.D.

Academic Editor

PLOS Genetics

Geraldine Butler

Section Editor

PLOS Genetics

Aimée Dudley

Editor-in-Chief

PLOS Genetics

Anne Goriely

Editor-in-Chief

PLOS Genetics

Comments from the reviewers (if applicable):

The revised manuscript was reviewed by two of the original reviewers and a new reviewer (Reviewer #4). While all three reviewers are enthusiastic about this version of the manuscript, they have identified specific concerns in their comments. Please address these concerns in the final version of the manuscript and explain any changes made in an accompanying document.

Reviewer's Responses to Questions

**Comments to the Authors:**

Reviewer #2: Line 92: 'distinguished' instead of 'differentiated'

Figure 1B: I appreciate the additional info regarding the sorting. I, however, still find it difficult to assess, which strain now retained which origin and how long the mtDNAs are. A linear representation would in my view make the figure much more digestible and informative. Also, the strains could be labeled with e. g. IIa10 and 13, which would make it possible to relate them to other figures.

Line 138-139: Why is it "the previously isolated..."? Shouldn't it be rather "a previously isolated..."? "the" implies that the reader should know the strain.

Figure 1E: Typo in y-axis label, incubation

Figure 2B: Can the authors describe the results obtained in relation to Figure 2B in a little more detail. What are the regions that have been additionally lost? Why do the subclones 2 and 3 for Ia14 show similar suppressivity, but differ in the coverage? Why has subclone 3 a higher suppressivity but is virtually indistinguishable from the other strains (except subclone 2)?

Line 222: What does not "carrying additional mutations in their genomic DNA" mean? Do the authors refer to auxotrophy marker?

Line 426-429: This did not change since the initial submission, but could the authors state a reason, how shortage of replication components lead to an advantage of hypersuppressive genomes?

Reviewer #3: The authors have largely addressed my concerns in the current revision, and I believe that the updated manuscript is appropriate for publication in PLoS Genetics. The authors may wish to consider the following minor points.

Minor points.

Figure 1, Page 9. One of the author’s points is that the size of the mitochondrial genome is not correlated with either its suppressivity or copy number. It might be worth noting that the strongest suppressors have biased retention of the mitochondrial genome (Fig 1B). They appear to retain ORI2 and ORI7 but not include ORI6. Another observation that the authors currently don’t make is that no single region of the mitochondrial genome is retained in all rho- isolates (consistent with nuclear encoding of the mitochondrial DNA polymerase).

Figure S1. Presumably the read counts plotted come from a rho+ strain. The authors should update the figure legend to indicate this.

Figure S6, Page 11. The authors should modify their description of their ad hoc assessment of the consistency of a trend towards increasing suppressivity. The authors state that “the initial suppressivity assessment was on average lower (5.9% +/- 10.9%) than subsequent ones”. Although true, the authors should add a statement to the effect that despite this general trend, there were isolates whose initial assessments were higher than the subsequent ones.

Reviewer #4: I have read the revised version of your manuscript and examined your answers to the questions of the three previous reviewers with whom I agree.

As I understand your work was made with spontaneously arised rho- mutants (not EtBr induced) from strain W303 but their sequences were mapped to the reference S288c (line 502). Why ? This can only ignore the polymorphic variations between yeast strains, which is high, especially for mtDNA.

Second, as I also understand you measure suppressiveness (following its historical discovery) by plating zygotes (or early zygotic buds) onto synthetic medium, differential for respiration. This is fine but remember that in that case, petite colonies are those in which 100% of mtDNA molecules are mutated (otherwise respiratory competent papillae would appear). And the fact that such colonies are respiratory deficient do not prove that theit mtDNA is identical to the parental rho- mutant tested. Neither that there is only one type of mtDNA molecules in each clone. This point was raised by reviewer 3 and is critical. The fact that you have yourself rediscovered that mtDNA molecules of some rho- mutants are mitotically unstable and can generate other deletions strengthen this point.

Your discussions about origines of replication (ORI) and mitochondrial inhertiance in vertebrates are interesting but not exactly address your point. It is now known since many years that yeast mtDNA molecules without any ORI sequence replicate well, and can even show some degree of suppressiveness. And mtDNA molecules of vetebrates are very differet from those of yeasts.

**Have all data underlying the figures and results presented in the manuscript been provided?**

Reviewer #2: Yes

Reviewer #3: Yes

Reviewer #4: Yes

PLOS authors have the option to publish the peer review history of their article (what does this mean? ). If published, this will include your full peer review and any attached files.

**Do you want your identity to be public for this peer review?** For information about this choice, including consent withdrawal, please see our Privacy Policy .

Reviewer #2: No

Reviewer #3: No

Reviewer #4: No

**Data Deposition**

http://datadryad.org/submit?journalID=pgenetics&manu=PGENETICS-D-24-01121R1

**Press Queries**

---

## [Editor Report · Acceptance letter]

PGENETICS-D-24-01121R1

Inheritance bias of deletion-harbouring mtDNA in yeast: the role of copy number and intracellular selection

Dear Dr Knorre,

We are pleased to inform you that your manuscript entitled "Inheritance bias of deletion-harbouring mtDNA in yeast: the role of copy number and intracellular selection" has been formally accepted for publication in PLOS Genetics! Your manuscript is now with our production department and you will be notified of the publication date in due course.

With kind regards,

Zsofia Freund

PLOS Genetics

On behalf of:
